# PRRSV Elimination in a Farrow-to-Finish Pig Herd Using Herd Closure and Rollover Approach

**DOI:** 10.3390/v15061239

**Published:** 2023-05-25

**Authors:** Ruiming Hu, Tiansheng Zhang, Rongbin Lai, Zhen Ding, Yu Zhuang, Hao Liu, Huabin Cao, Xiaona Gao, Junrong Luo, Zheng Chen, Caiying Zhang, Ping Liu, Xiaoquan Guo, Guoliang Hu, Nengshui Ding, Shunzhou Deng

**Affiliations:** 1Department of Veterinary Medicine, College of Animal Science and Technology, Jiangxi Agricultural University, No. 1101 Zhimin Avenue, Nanchang 330045, China; rhu@jxau.edu.cn (R.H.); chbin20020804@jxau.edu.cn (H.C.); chenzheng@jxau.edu.cn (Z.C.); zhangcaiying0916@jxau.edu.cn (C.Z.); hgljx3818@jxau.edu.cn (G.H.); 2Jiangxi Provincial Key Laboratory for Animal Disease Diagnosis and Control, Institute of Animal Population Health, Jiangxi Agricultural University, Nanchang 330045, China; 3Key Laboratory of Swine Nutrition and Feed Science of Fujian Province, Aonong Group, Zhangzhou 363000, China; 4State Key Laboratory for Pig Genetic Improvement and Production Technology, Jiangxi Agricultural University, Nanchang 330045, China

**Keywords:** PRRSV, elimination, farrow-to-finish

## Abstract

It is well established that PRRSV elimination is an effective strategy for PRRS control, but published reports concerning successful PRRSV elimination cases in farrow-to-finishing herds are rare. Here, we have reported a successful PRRSV elimination case in a farrow-to-finish herd by employing a “herd closure and rollover” approach with some modifications. Briefly, the introduction of pigs to the herd was stopped and normal production processes were maintained until the herd reached a PRRSV provisional negative status. During the herd closure, strict biosecurity protocols were implemented to prevent transmission between nursery pigs and sows. In the current case, introducing gilts before herd closure and live PRRSV exposure were skipped. In the 23rd week post-outbreak, the pre-weaning piglets started to show 100% PRRSV negativity in qPCR tests. In the 27th week, nursery and fattening barns fully launched depopulation. In the 28th week, nursery and fattening houses reopened and sentinel gilts were introduced into gestation barns. Sixty days post-sentinel gilt introduction, the sentinel pigs maintained being PRRSV antibody negative, manifesting that the herd matched the standard of the provisional negative status. The production performance of the herd took 5 months to bounce back to normal. Overall, the current study provided additional information for PRRSV elimination in farrow-to-finish pig herds.

## 1. Introduction

Porcine reproductive and respiratory syndrome virus (PRRSV), the causative agent of porcine reproductive and respiratory syndrome (PRRS), is an enveloped single-stranded positive-sense RNA virus, classified to the order of Nidovirales, the family Arteriviridae, and the genus Arterivirus [1,2]. PRRSV mainly leads to sow reproduction failure, respiratory symptoms, increased mortality, and a lower feed conversion ratio. In pig herds, PRRS can maintain a long-lasting endemic status, leading to enduring economic losses [2].

Since PRRS has emerged in the last century, PRRS is endemic in most countries worldwide, causing tremendous economic losses [3]. In 2013, one study estimated that PRRS-associated diseases costed approximately USD 660 million per year in the United States [4]. China has the world’s largest pig production scale with approximately 40–45 million sows in stock, and it produces 600–700 million pigs every year [5,6,7]. Therefore, it is rational to speculate that PRRS has led to enormous economic losses in China’s pig industry [7].

A standardized terminology to define and communicate the PRRSV status of breeding herds, proposed by Holtkamp in 2021, was adopted in this study. In short, “positive unstable (I)” represents the breeding herds showing detectable PRRSV in weaning pigs via qPCR. As PRRSV is controlled and eventually eliminated, the herds move to a “positive stable (II-A/II-vx)” status, followed by a “provisional negative (III)” and ultimately a “negative (IV)” status. To match the provisional negative status, more than 60 sentinel gilts must maintain 100% negative PRRSV antibody for more than 60 days, demonstrating that PRRSV is not present in the herd. To reach a “negative status (IV)”, all of the animals in the herd have to be PRRSV negative via a PRRSV antibody test [8,9]. In addition, if the herd applies PRRSV elimination via herd closure and rollover, to match the standard of a “negative status”, the removal of all previously infected animals from the herd is required [8,9].

PRRSV control is difficult for most pig herds. Various strategies and efforts have been applied for PRRS control. Vaccination is the most common choice for viral pathogen control; however, the efficiency of the PRRSV-modified live vaccine (MLV) is sometimes inconsistent and controversial. There are several commercial PRRS MLVs available in China, containing various PRRSV strains such as VR2332 (Lineage 5), JXA1-R (Lineage 8.7), HuN4-F114 (Lineage 8.7), and TJM-F92 (Lineage 8.7), while no vaccine strain belongs to the NADC30- or NADC34-like strains. In fact, PRRSV is notorious for the constant changes in pathogenesis and antigenic profiles [1,10,11,12]. According to the epidemic investigations, the dominant PRRSV field strains showed significant alterations every 8–10 years in endemic areas or countries with substantial antigen drift [1,11,13]. Pig herds are highly vulnerable toward newly emerged PRRSV genotypes, such as JXA1-like strains which emerged in China in 2006 and NADC30-like strains which emerged in the United States in 2008 [1,11,13].

Consequently, choosing an appropriate MLV for PRRSV control can be challenging. While levels of genetic homology (mostly only the homology of ORF5) between vaccine strains and field strains have often been used as an indicator of vaccine efficacy [14], the protection efficacy of a vaccine against a certain field virus is not always directly linked to the level of sequence homology that it shares with the challenging strain [15]. Additionally, PRRS MLV has raised safety concerns due to the potential for virulence enhancement and recombination between vaccine strains and field strains [16,17]. Furthermore, biosecurity strategies such as the McRebel rule, segregated production system, early weaning, all-in all-out system, and gilt acclimation have also been applied for PRRSV control and have achieved positive effects [18,19].

It is well established that PRRSV elimination is a powerful strategy to restore a high production level in pig farms. There are three PRRSV elimination strategies applied in the field: (1) whole herd depopulation and repopulation; (2) test and removal; and (3) herd closure and rollover (also called load–close–exposure, LCE) [20,21,22]. The depopulation strategy shows the highest efficiency and lowest risk but the cost is too high for most pig herds. The test and removal strategy is not the common choice because of the high cost of qPCR tests and it being time consuming. Only the last one shows an acceptable cost and has become the most popular strategy for PRRSV elimination [23].

The herd closure and rollover strategy typically involves three steps: (1) Introducing the PRRSV-negative gilts in one batch. To minimize the cost, the amount of PRRSV-negative gilts should be enough for the sow replacement during herd closure; (2) Herd closure until the herd reaches a provisional negative status, which usually takes 210–250 days; (3) Expose all of the animals to the PRRSV live virus at the same time [22,24,25,26]. The epidemiological basis of this strategy is that if the sow population is infected by PRRSV at the same time and recovers simultaneously, the entire population will establish sterilizing immunity against field strains. Since the sensitive host is eliminated and no pigs are introduced during the herd closure, the field live viruses will eventually be eliminated from the herds.

To make sure that all of the sows and gilts are exposed simultaneously, natural exposure may be replaced with simultaneous vaccination of MLV or inoculation with serum containing resident virus [22,24,25,26]. Yet, deliberate exposure is still a controversial step. Deliberately exposing pigs to a field live virus (FLV) raises serious biosecurity concerns and it may substantially increase the mortality rate, especially in younger gilts [22,27]. Additionally, the effectiveness of MLV is inconsistent [17]. Actually, the dominant field strains in China are the NADC30- and NADC34-like strains, which show significant antigen drift compared with all of the vaccine strains in China [6,28,29]. In addition, the live PRRSV vaccine may show virulence enhancement during cycling in the herd, which also raises serious concerns [7,16,17,30]. In this condition, if the herd is capable of establishing sterilizing immunity against the field strain without deliberate exposure, it may be preferable to skip this step.

It is worth noting that farrow-to-finishing facilities are still the dominant production system in many countries, including China. In these kinds of facilities, nursery and fattening pig depopulation should be launched before herd closure, and all of the piglets should be sold or moved off the pig farm after weaning [27]. However, shutting down the nursery barns significantly increases costs, which is a major concern for pig producers. Thus, how to minimize the period of nursery barns shutting down is a key point to reduce the costs of the PRRSV elimination program [27,31].

Over the past decade, the “herd closure and rollover” strategy has been introduced and adopted in Chinese pig herds. One study showed four different PRRSV elimination protocols in four different pig farms in China using the “load–close–exposure” approach [23]. All of the farms used the MLV vaccination in a sow population, while two out of four farms used FLV exposure in gilts introduced before exposure. Their results showed that both MLV and FLV exposure were effective, although FLV exposure led to higher mortality in gilts. Interestingly, the endemic field strains of four pig farms belonged to four different lineages [32]. The pig farms took 21–22 weeks to produce PRRSV-negative pre-weaning pigs and 5–6 months to achieve the baseline production level [23]. However, only a few reports have disclosed the elimination processes and shared experiences of elimination protocol design in China [23]. The aim of this study was to demonstrate how an elimination approach was applied in a farrow-to-finishing herd after a severe PRRSV outbreak by applying the herd closure and rollover approach without deliberate live virus exposure. Our experience may help more pig producers with farrow-to-finishing facilities to design and optimize their elimination protocols in the future.

## 2. Methods

### 2.1. PRRSV Viral RNA and Antibody Detection

The ELISA kit PRRS 3X ab (IDEXX, Westbrook, ME, USA) was used for PRRSV antibody detection. Total RNA extraction was applied using an RNeasy Micro Kit (Qiagen, Hilden, Germany). The qPCR kit RealPCR PRRS (IDEXX, Westbrook, ME, USA) was used for PRRSV viral RNA detection.

### 2.2. PRRSV Elimination Protocol

The producer wished to eliminate PRRSV from the herd, so that they could resume the sale of PRRSV-antibody-negative gilts from the herd as soon as possible. In this circumstance, a PRRS elimination program was designed and started in September 2020. The timeline of the PRRSV elimination process is shown in Figure 1. In the current study, the deliberate exposure of live viruses was skipped. Additionally, no gilts were introduced after the PRRSV outbreak.

The herd closure began on 22 August 2020, which was the 15th day (3rd week) post-PRRSV outbreak (Figure 1). Gilt introduction was delayed until the herd reached a provisional negative status. During herd closure, all semen was purchased from outside suppliers. Each batch of semen proved to be negative of common pathogens via qPCR tests, including PEDV, PRRSV, PCV2, PRV, ASFV, CSFV, and so on. Piglet production continued without interruption. Nursery and fattening houses maintained regular production processes before depopulation, which was launched in the 27th week after the pre-weaning piglets tested negative for PRRSV for 4 consecutive weeks (Figure 1). During herd closure, a modified McREBEL protocol was implemented and strict internal biosecurity measures were applied to limit PRRSV transmission within the farm [3]. The farm was divided into several independent regions, including breeding (gestation and farrowing barns), nursery and fattening, gilt development, waste collection and processing, and staff living areas. The flow of people and materials was restricted within each region and potential cross-overs were eliminated. The sanitization frequency was doubled during herd closure. The flow of pigs and manure was strictly maintained in a one-directional movement with no turning back. Sows exhibiting PRRS-associated symptoms such as fever, abortion, or returning to estrus were moved to a segregated region for PRRSV viral RNA detection.

In the current study, the nursery and fattening houses started full depopulation after 4 uninterrupted weeks of the pre-weaning pigs showing 100% PRRSV negativity, followed by thorough cleaning and sterilization. Following this, the nursery and fattening houses reopened again to accept the PRRSV-negative weaned piglets from the farrowing houses. Meanwhile, 90 PRRSV-naïve gilts were introduced into gestation barns, neighboring with sows as sentinel pigs. Afterward, sentinel pigs were introduced, and both pre-weaning pigs and sentinel gilts were monitored via qPCR and ELISA tests monthly. If no sentinel pig shows positive in both qPCR and ELISA tests for at least 60 days after introduction, the herd reaches a provisional negative status.

### 2.3. PRRSV Diagnosis and Surveillance Program

In the beginning of the PRRS outbreak, 10 samples (5 blood samples from sows and 5 tissue samples from stillbirth fetuses) were collected for qPCR and ELISA tests, and no sample was pooled. The positive samples were subjected to ORF5 sequencing and phylogenetic analysis. In the 2nd week post-PRRS outbreak, to investigate the infection ratio of PRRSV, 65 blood samples were collected from the sows with symptoms for qPCR and antibody tests.

During the herd closure, PRRSV surveillance started in the 15th week post-PRRSV outbreak. The testicle processing fluids of the whole litter were pooled as 1 sample and every litter was sampled for qPCR tests from the 15th to 62nd week post-outbreak. The mating schedule was designed as one batch per week; hence, the test frequency of the testicle processing fluids was weekly. Furthermore, antibody surveillance was carried out to monitor the dynamic of PRRSV circling in the herd. For antibody surveillance, 30 sera samples of sows and gilts, and 50 sera samples from weaning piglets (21 to 25 days old) were collected in the 5th, 14th, 26th, 38th, 50th, and 62nd week post-PRRSV outbreak.

After the introduction of sentinel pigs, blood samples of every sentinel pig were collected for qPCR and ELISA tests on the 30th and 60th day post-introduction. When the nursery and fattening houses reponed after depopulation, blood samples and the oral fluids of nursery pigs over 60 days old were collected monthly for qPCR and ELISA tests. Serum samples were used for ELISA tests, while oral fluids were used for qPCR tests (the oral fluids from one barn were combined as 1 sample). The sampling ratio of both blood samples and the oral fluids in the nursery and fattening houses was 10% from the 28th to 62nd week post-outbreak.

After the 62nd week post-outbreak, the surveillance frequency of both the antibody and qPCR tests was decreased to semi-annually. For the qPCR test in pre-weaning piglets, the testicle processing fluids of the whole litter were pooled as 1 sample and the sampling ratio was 20%. For antibody surveillance post-62nd week, the sera samples were randomly collected from sows, weaning piglets, nursery pigs, fattening pigs, and gilts. Each production stage collected 30 sera samples per time.

### 2.4. Production Data Collection

The major data concerning the production level were acquired from the management system adopted in the farms, including (1) mating rate after weaning (7 days); (2) the farrowing rate of gestation sows per month; (3) abortion rate per month; (4) stillbirth/litter; (5) piglets born alive per litter; (6) mortality rate of suckling piglets; (7) average weight after weaning; and (8) mortality rate of nursery piglets. All data include the 6 months before the outbreak and the 17 months post-outbreak.

## 3. Results

### 3.1. The Herd Background

In the current study, the pig farm was a great grandparent (GGP) herd. The herd was in a farrow-to-finish facility, containing 1252 sows and 630 gilts, with corresponding nursery houses, fattening houses, and a gilt development unit. On 11 August 2020, a PRRSV outbreak took place in this herd (Figure 1). Before the outbreak, the herd had maintained a PRRSV-antibody-negative status since the production was initiated and the PRRSV live vaccine had never been used in this herd.

Overall, most endemic diseases had been well controlled in this pig farm from 2020 to 2022. Before the PRRSV outbreak, the herd was free from the pseudorabies virus and PRRSV. Additionally, the porcine epidemic diarrhea virus (PEDV) and classic swine fever virus (CSFV) had never been detected via qPCR in the herd. Mycoplasma hyopneumoniae and PCV2 are endemic in most pig farms. However, the clinical signs associated with both pathogens had seldomly been observed; thus, the farm did not monitor both pathogens.

### 3.2. The Impact of PRRSV Outbreak

During the outbreak, the sows exhibited severe reproductive failure (Figure 2A). The mating rates after weaning (7 days) and the farrowing rate dropped to 54.78% and 59.05% post-outbreak, respectively, while the abortion rate and stillbirth rate per litter raised to 43.5% and 7.76, respectively (Figure 2A). For the suckling piglets, the piglets born alive per litter decreased from 13.7 to 7.97, and the weaned piglets per litter decreased from 12.99 to 6.97, while the weaning weight decreased from 6.38 kg to 5.14 kg (Figure 2B). Furthermore, the mortality rate of the suckling piglets and nursery piglets increased to 27.97% and 25.67% post-outbreak, respectively (Figure 2B). These data indicate that PRRSV spread to the whole population in a short period of time and caused severe economic losses.

### 3.3. Diagnostic Results

Within 3 days post-outbreak, 10 clinical samples were collected (5 blood samples from sows showing clinical signs and 5 tissue samples of stillbirth fetuses) for both qPCR and ELISA tests. The average S:P value of the serum was 1.47, ranging from 0.82 to 1.91, indicating that robust PRRSV infection was circling in the sow population. The qPCR results showed 100% positivity for PRRSV. The other potential pathogens were all negative, including PRV, CSFV, and ASFV. PCV2 was positive in two tissue samples, suggesting it was not the causative pathogen. The ELISA and qPCR results both confirmed that the outbreak was caused by PRRSV infection. The open reading frame 5 (ORF5) of the field strain was sequenced. Multi-sequence alignment and phylogenetic analysis indicated that the field strain was a lineage 1.5 JXA1-like strain (Figure 3).

In the 2nd week post-outbreak, a sera epidemiological investigation was conducted to investigate the infection ratio of the sow population within the population. The ELISA results showed 100% positivity, and the average S/P value was 1.87, ranging from 0.87 to 2.63. These results indicated that the whole sow population had been infected.

Based on the diagnostic results, the PRRSV elimination protocol was designed and carried out on the 15th day (3rd week) post-outbreak. The PRRSV elimination timeline is shown in Figure 1.

### 3.4. The Time to Achieve Provisional Negative Status

As shown in Figure 1, the first day of the PRRSV outbreak was set as Day 1. In the 3rd week post-PRRS outbreak, the herd closure was initiated using the protocol described in the Methods section. Starting in the 15th week post-outbreak, PRRSV surveillance was carried out. For the pre-weaning pigs, testicle processing fluids were collected and subjected to qPCR tests weekly using a commercial qPCR kit (IDEXX, Westbrook, ME, USA). As shown in Table 1, the PRRSV-positive ratio of the pre-weaning pigs declined quickly and became 100% negative by 23 weeks post-outbreak. In the 27th week, the nursery and fattening pigs were depopulated. In the 28th week, the PRRSV-negative gilts were introduced into gestation barns as sentinel pigs and the nursery house started to receive the weaning pigs. ELISA and qPCR tests demonstrated that no sentinel gilts showed seroconversion or positive results in qPCR tests in the following 9 weeks. Additionally, the nursery pigs maintained PRRSV negativity in qPCR tests in the following 9 weeks after the reopening. These results indicated that the herd had achieved the standard of a provisional negative status. In the 37th week, the herd closure finished and PRRSV-naïve gilts were introduced to replace the sows experiencing the PRRSV outbreak step by step. Overall, the herd took 21 weeks to match the status of time-to-PRRSV-stability (TTS) and 35 weeks to meet the standard of a provisional negative status. 

The PRRSV antibody was detected via PRRS 3X Ab (IDEXX, Westbrook, ME, USA), as shown in Table 2. The antibody level of sows and weaning piglets kept declining during herd closure. In the 38th week post-outbreak, 1 week after the herd reached a provisional negative status, 33.3% of samples of sows and gilts showed a PRRSV-antibody-positive result, while only 4% of pre-weaning piglets were PRRSV antibody positive, indicating that the herd was close to producing PRRSV-negative offspring confirmed via both PRRSV ELISA and viral qPCR tests (Table 2). The pre-weaning pigs started to show 100% antibody negativity in the 50th week post-outbreak, while the sow population turned to being antibody negative in the 62nd week post-outbreak. These results suggested that PRRSV circling faded quickly in the herd.

All of the sows and gilts that experienced the outbreak had been replaced by PRRSV-naïve gilts by March 2022, which was the 74th week post-PRRSV outbreak. The surveillance program continued until June 2022, with semi-annually test frequency. In the end, both qPCR and ELISA results were all negative for the pre-weaning piglets, sows, nursery pigs, and gilts, indicating that the elimination program had succeeded.

### 3.5. The Time of Production Backing to Baseline

PRRS outbreak or endemic could significantly undermine production efficacy and cause substantial economic loss. In the current case, the statistical data concerning the reproductive efficacy of sows, including mating rate, farrowing rate, abortion, and stillbirth, returned back to the baseline in February 2021, which happened 5 months post-herd closure (Figure 2A). For suckling piglets, the average live born piglets per litter, the piglets weaned per litter, and the mortality rate of suckling piglets returned back to the level before the outbreak in January 2021, which took 4 months post-herd closure (Figure 2B). The average weaning weight bounced back to baseline in October 2020, which may partially have resulted from a lower number of piglets per litter. For the nursery piglets, the mortality rate increased to 25.67% during the outbreak and dropped to the baseline quickly in November 2020. Overall, within 5 months post-herd closure, the production performance was fully back to the level before the outbreak.

## 4. Discussion

A PRRSV endemic in pig farms causes sow reproduction failure, promotes secondary infection, increases the mortality rate, and impairs the feed-to-meat ratio [3]. More importantly, it is difficult to effectively terminate PRRSV endemicity, leading to sustained economic losses. Some pig herds can maintain a PRRSV-positive stable status (II-A) through modified live virus (MLV) vaccination [20,33,34,35], but the effectiveness of MLV is inconsistent and controversial [17]. Additionally, for GGP herds, producing PRRSV-negative replacing females is critical for the business. In the last decade, the traditional back-yard pig farms have been rapidly replaced by large-scale modernized pig farms, leading to an increase in attempts to eliminate PRRSV in China.

For a PRRS elimination program, the load–close–exposure strategy is the most common choice in most conditions [3,36]. To make sure that every individual of the sow population is exposed simultaneously, natural exposure can be replaced by simultaneous vaccination of MLV or by inoculating pigs with serum containing a live resident virus [25]. However, the deliberate exposure to a live virus raises several concerns. Primarily, deliberately exposing pigs to a field live virus (FLV) is a controversial step, because this step may lead to increased mortality, biosecurity issues, field virus dissemination, and spillover [23,37]. In addition, the effectiveness of MLV vaccination is inconsistent and controversial because of antigen drift, which happens frequently [15]. Additionally, PRRSV live vaccines have also exhibited virulence enhancement during field transmission [12,16,17]. More importantly, introducing another PRRSV live strain into a herd with robust field strain circulation may lead to a potential risk of recombination between two different strains and the rapid evolution of the field strain, resulting in unpredictable and complicated scenarios [16,17]. Several studies have reported on recombinant PRRSV strains such as GM2 and QYYZ, possibly resulting from the recombination between field strains and vaccine strains [29,38,39].

In the current study, herd closure was applied without exposure to a PRRSV live virus for several reasons. Firstly, the herd was a PRRSV-naïve population that had just experienced a severe PRRSV outbreak. An epidemiological investigation showed that the PRRSV infection ratio reached 100% within 2 weeks of the outbreak. In this condition, deliberate exposure to FLV was unnecessary. Secondly, vaccination with a live vaccine was also excluded because all sows and gilts had already been infected with the field strain. The third reason was concern over potential virulence enhancement in the JXA1-like live vaccine, which has been reported previously [16]. Our results suggest that a herd can still establish sterilizing immunity in a short period of time without deliberate exposure. Once we had decided to skip deliberate exposure, introducing more PRRSV-naïve gilts during the PRRSV outbreak would have further enhanced PRRSV circulation, which would have been inappropriate. Therefore, gilt introduction before herd closure was also skipped in this case.

To avoid the potential risks associated with deliberate exposure to either FLV or MLV, Attila Pertich et al. reported a PRRSV elimination program that combined inactivated vaccination with a rollover method in a Hungarian pig herd [22]. Their results showed that by immunizing the pigs with an inactivated vaccine, PRRSV-negative offspring were successfully produced and raised in another facility as gilts. Once enough PRRSV-negative gilts were available to replace the entire sow population, sow depopulation was initiated and a PRRSV-negative population was successfully established [22]. This is an exciting result that warrants further investigation. However, the effectiveness of inactivated or subunit vaccines against PRRSV is still unclear and it is uncertain whether this case is replicable.

According to DCL Linhares’s study, which investigated the effectiveness of the load–close–exposure approach, the time to reach TTS ranged from 12 to 42 weeks among 61 participating herds with a median of 26.6 weeks [24]. Another study summarized a PRRS elimination program in four pig farms in China. Their results showed that TTS ranged from 21 to 22 weeks [23]. Both studies mentioned above applied deliberate exposure. Usually, if deliberate exposure of the live virus is not applied, the TTS will be longer. Interestingly, in the current study, the TTS was 21 weeks post-herd closure, shorter than the median number (26.6 weeks) in Linhare’s study [24]. We speculate that the natural PRRSV infection was quite severe in the current case because all of the sows and gilts were PRRSV-naïve before the outbreak. Consequently, the natural exposure also achieved sterilizing herd immunity in the whole population in the same period. This comparison suggests that deliberate exposure may not always be necessary. However, in PRRSV endemic herds, deliberate exposure to a field strain or MLV would be necessary for achieving sterilizing herd immunity in a short period of time.

In the current study, the herd took 22 weeks (5 months) to return to the pre-outbreak production levels. According to Dr. Linhares’s study, the overall median time to baseline production (TTBP) was 16.5 weeks and ranged from 0 to 29 weeks [24]. The TTBP is clearly dependent on various factors, including the overall health level of the sows, the virulence of the PRRSV field strains, and the age structure of the sow population. Among these factors, the most essential is the health level of the sow population. Our observations suggest that if the sow population is affected by more than just PRRSV or if secondary infections are severe, the recovery period will be longer than average. If the proportion of gilts is relatively large, the recovery period will also be longer.

It is worth noticing that the pig farm in this study was a farrow-to-finish herd, which is still quite common in China. Theoretically, nursery and fattening pig depopulation should be initiated at the beginning of herd closure. According to the producer’s estimate, if nursery and fattening houses were closed at the beginning, the extra cost would have been CNY 14,000 per litter. During herd closure, a total of 2482 litters were produced, resulting in an extra cost of CNY 34.7 million (equivalent to USD 5.3 million) in total. For pig producers, the biggest concern with PRRSV elimination is costs. To reduce costs, production in nursery and fattening houses was maintained until 27 weeks post-outbreak. The delayed depopulation of the nursery and fattening houses obviously increased potential risks. In this case, a strict internal biosecurity protocol was applied to prevent PRRSV circulation between the sows and nursery pigs. Ultimately, the herd closure period was not significantly extended, demonstrating that the biosecurity protocol effectively blocked PRRSV circulation between different production areas.

According to our observations, several key points were critical for minimizing the risk of delayed depopulation in this case: (1) strict biosecurity protocols must be followed by staff; (2) the immune response level established in the sow population must be high enough to prevent reinfection in a short period of time; and (3) the PRRSV-positive rate in pre-weaning pigs must decline quickly.

## 5. Conclusions

Overall, this study has described a PRRSV elimination program in a farrow-to-finish herd with minimal costs and a relatively short herd closure period. Our experience demonstrates that deliberate exposure is not always necessary when PRRSV has just broken out in a herd. Additionally, depopulation of nursery houses can be delayed in farrow-to-finish facilities, which significantly reduces elimination costs.

## Figures and Tables

**Figure 1 viruses-15-01239-f001:**
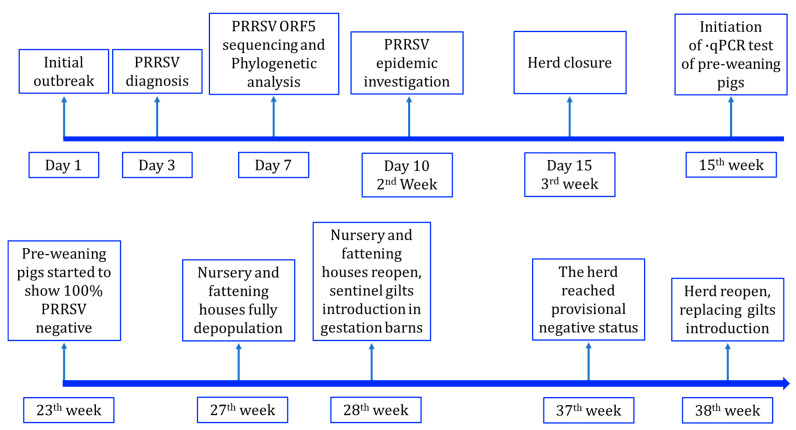
Timeline of PRRSV elimination program in current study.

**Figure 2 viruses-15-01239-f002:**
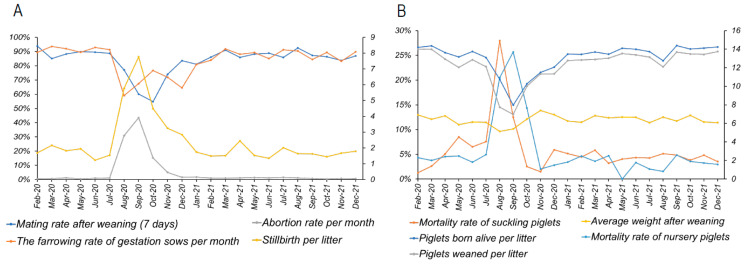
Production data of the herd. The production data were extracted from the management system adopted in the farms from February 2020 to December 2021. (**A**) Mating rate after weaning (7 days), monthly farrowing rate of gestation sows, and monthly abortion rate are shown on left y axis, while stillbirths per litter are shown on the right y-axis. (**B**) The mortality rates of nursery piglets and suckling piglets are shown on the left y-axis, while piglets born alive per litter, piglets weaned per litter, and average weight after weaning are shown on the right y-axis. All data include the 6 months before the outbreak and the 17 months post-outbreak.

**Figure 3 viruses-15-01239-f003:**
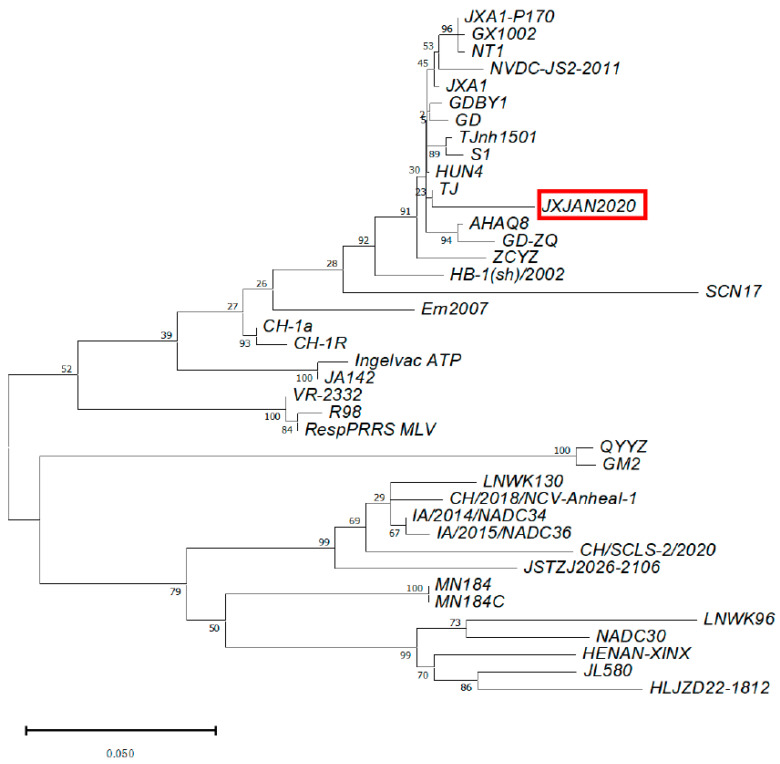
Phylogenetic analysis of field PRRSV strain JXAN2020. The blood samples of sows showing typical PRRS symptoms were collected and subjected to PCR amplification of ORF5 followed by Sanger sequencing. The ORF5 sequence was aligned with multiple reference sequences obtained from GenBank. A phylogenetic tree was constructed using the MEGA X package with the neighboring join algorithm. The red frame indicates the ORF5 sequence of the PRRSV field strain determined in the current study.

**Table 1 viruses-15-01239-t001:** PRRSV qPCR results of testicle processing fluids of pre-weaning pigs.

Sampling Time	qPCR Results	Positive Ratio	Sampling Time	qPCR Results	Positive Ratio	Sampling Time	qPCR Results	Positive Ratio
15th week *	3/52	5.77%	25th week	0/57	0.00%	35th week	0/48	0.00%
16th week	2/55	3.64%	26th week	0/52	0.00%	36th week	0/52	0.00%
17th week	3/49	6.12%	27th week	0/50	0.00%	37th week	0/51	0.00%
18th week	0/51	0.00%	28th week	0/55	0.00%	38th week	0/51	0.00%
19th week	2/53	3.77%	29th week	0/53	0.00%	42nd week	0/57	0.00%
20th week	1/55	1.82%	30th week	0/52	0.00%	46th week	0/56	0.00%
21st week	0/61	0.00%	31st week	0/52	0.00%	50th week	0/57	0.00%
22nd week	1/57	1.75%	32nd week	0/54	0.00%	54th week	0/49	0.00%
23rd week	0/50	0.00%	33rd week	0/63	0.00%	58th week	0/54	0.00%
24th week	0/51	0.00%	34th week	0/57	0.00%	62nd week	0/51	0.00%

*: The time indicates the weeks post-PRRSV outbreak.

**Table 2 viruses-15-01239-t002:** Distribution of PRRS ELISA S:P values in sows, gilts, and pre-weaning pigs.

Sample Collection Time	Sows and Gilts						Pre-Weaning Pigs					
	S:P Value		Distribution of S:P Values				S:P Value		Distribution of S:P Values			
	Mean	Range	<0.4	0.4–1.0	1.1–2.0	>2.0	Mean	Range	<0.4	0.4–1.0	1.1–2.0	>2.0
5th week *	1.97	0.87–2.83	0/30	3/30	9/30	16/30	1.66	0.22–2.37	6/30	11/30	9/30	4/30
14th week	1.713	0.43–1.8	1/30	9/30	24/30	0/30	0.67	0.24–1.84	16/50	21/50	8/50	5/50
26th week	0.78	0.23–1.7	11/30	17/30	5/30	0/30	0.49	0.16–0.87	42/50	8/50	0/50	0/50
38th week	0.35	0.28–1.1	20/30	7/30	3/30	0/30	0.26	0.08–0.52	48/50	2/50	0/50	0/50
50th week	0.31	0.31–0.89	29/30	1/30	0/30	0/30	0.21	0.03–0.37	50/50	0/50	0/50	0/50
62nd week	0.16	0.12–0.34	30/30	0/30	0/30	0/30	0.17	0.04–0.33	50/50	0/50	0/50	0/50

*: The time indicates the weeks post-PRRSV outbreak.

## Data Availability

The datasets generated for this study are available on request to the corresponding author.

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
