# Peer review of "PRRSV Elimination in a Farrow-to-Finish Pig Herd Using Herd Closure and Rollover Approach"

_viruses, 2023, doi:10.3390/v15061239_

Round 1

Reviewer 1 Report

Comments and Suggestions for Authors

Congratulations on your submission. Your paper describes a case study of a PRRSV elimination program performed in one wean-to-finish farm in China.

The authors need to add more details in the methods section so readers can evaluate the internal and external validity of your findings. 

Comments:

- I have provided a few comments in the attached file.

 - Provide more details about the status of the farm for PRRSV and other endemic diseases such as PEDV, Mhyo, etc.

- Overall, the authors must include more details on how the herd closure and exposure were performed (number of FVI exposure).

- Better description of the surveillance scheme to match with table 2.

Regarding the AASV classification, there is a new proposed classification for PRRSV virus - Holtkamp D, Torremorell M, Corzo CA, Linhares DCL, Almeida MN, Yeske P, Polson DD, Becton L, Snelson H, Donovan T, Pittman J, Johnson C, Vilalta C, Silva GS, Sanhueza J. Proposed modifications to porcine reproductive and respiratory syndrome virus herd classification. J Swine Health Prod. 2021;29(5):261-270.

Comments on the Quality of English Language

English is good, I didn't see major issues with English.

Reviewer 2 Report

Comments and Suggestions for Authors

The manuscript viruses-2388594 consists of the description of the successful PRRSV elimination in a farrow-to-finish herd by employing the so called “herd closure and rollover” protocol , as initially  described in the US by Torremorell et al.

The authors of the manuscript are correct when they state that, although the methodology they describe is widely used in a global context, successful cases of PRRSV elimination like the one they report are rare. In addition, this case corresponds to a successful application of the herd closure protocol in a Chinese farrow-to-finish farm involving close to 2,000 females. Such aspect of this report adds interesting comparative points when one considers that not all the technical steps and epidemiological diagnostic technologies used by these authors respond to those most used technical tools used in other countries like in North America.

The paper is well written and establishes logical analysis of results and conclusions based on adequate bibliographic references on the topic. Special emphasis is given by the authors to two points on their plan: 1) their avoidance of using planned infection using the autochthonous PRRSV strain isolated in the farm, which is always a controversial extreme procedure that may induce significant economic losses before the final successful output is reached. 2) their reluctance to use the most effective type of commercial vaccine known to date (MLV) to facilitate and uniformize the induction of massive protective immunity in the entire herd that is being closed. While I  as a reviewer, might dismiss as overcautious concerns by the authors to avoid the use an efficacious wisely applied vaccine, I do admit, however,  that such concerns stated by the authors may respond in this particular case to possible general uncertainty respect the commercial attenuated strains circulating in China.

Just one observation that I find it important should be corrected in this paper: The authors wrongly refer to the infectious PRRSV monitoring during the project as “antigen detection” as the single etiologic diagnosis used to trace the circulation of infectious PRRSV in the farm. Instead they just used PCR instead (RT PCR for PRRSV RNA detection, not antigen). (please correct on lines 54, 122, 145, 247 etc)

Comments on the Quality of English Language

no comments

Round 2

Reviewer 1 Report

Comments and Suggestions for Authors

Congratulations